# MIESRA mHealth: Marital satisfaction during pregnancy

Besral Besral[1], Misrawati Misrawati[2,3]*, Yati Afiyanti[2], Raden Irawati Ismail[4‡], Hidayat Arifin[5,6‡]

1 Department of Biostatistics, Faculty of Public Health, Universitas Indonesia, Depok, Indonesia,
2 Department of Maternity and Women Health, Faculty of Nursing, Universitas Indonesia, Depok, Indonesia,
3 Department of Maternity and Women Health, Faculty of Nursing, Universitas Riau, Pekanbaru, Indonesia,
4 Department of Psychiatrics, Faculty of Medicine, Universitas Indonesia, Depok, Indonesia, 5 Department of Fundamental Nursing Care, Faculty of Nursing, Universitas Airlangga, Surabaya, Indonesia, 6 School of Nursing, College of Nursing, Taipei Medical University, Taipei, Taiwan

☯ These authors contributed equally to this work.
‡ RII and HA also contributed equally to this work.
* misrawati@lecturer.unri.ac.id

## Abstract

The transition of a pregnant woman's role often causes emotional changes that have an impact on marital satisfaction. We develop MIESRA mHealth and evaluate its impact on satisfaction of husband-wife relationship during pregnancy. A quasi-experimental study was conducted on 82 couples of pregnant women and divided into control, single, and paired group. We implemented MIESRA mHealth for four weeks. In the couple group, the wife did mindfulness based on the information in the MIESRA mHealth together with her husband. In a single group, the wife sees the video as an initial guide to doing mindfulness. In the control group, respondents received programme interventions from hospitals which included education and consultation with obstetricians. Husband-wife relationship is evaluated using Compatibility of Husband-and-Wife Relationships / *Kesesuaian Hubungan Suami Istri* (KHSI) questionnaire and the generalised estimating equations (GEE) was used to analyse the data. The women's KHSI scores in the couple and single intervention groups (β = -7.46, p = 0.002; β = -9.11, p = 0.001) were better than the control group. The husbands' KHSI scores in the paired and individual intervention groups (β = -7.04, p<0.001; β = -3.74, p = 0.024) were better than the control group. Nursing interventions to build emotional bonds between parents and foetuses based on mHealth can be a promising intervention for marital harmony during the perinatal period. MIESRA m-Health is a promising intervention on marital satisfaction during pregnancy and can be implemented as a part of the antenatal care programme to increase marital satisfaction.

## Introduction

Pregnancy is a transition period in which women strive with biochemical, physiological and anatomical alterations [1]. Transition to motherhood is influenced by various factors across different levels, including individual factors (partner support, career aspirations),

**Data Availability Statement:** All relevant data are within the paper and its Supporting information files.

**Funding:** The authors received a grant from Kementerian Riset dan Teknologi/Badan Riset dan Inovasi Nasional with grant number: NKB-381/UN2.RST/HKP.05.00/2020 and grant recipient is Yati Afianti. The funders had no role in study design, data collection and analysis, decision to publish, or preparation of the manuscript.

**Competing interests:** The authors have declared that no competing interests exist.

organizational factors (family-friendly work practices, role models), and societal factors (social norms, attitudes towards the maternal body) [2]. Some pregnant women adjust well to these changes by maintaining optimal physical condition, positive interpersonal relationships, and gaining support from their family and health workers [3–5]. However, the transition is sometimes beyond control and makes pregnant women vulnerable to physical, mental, and social illnesses [6, 7]. Pregnant women mostly face psychological issues such as anxiety and depression during transition to motherhood [8, 9]. This condition can have an impact on miscarriage, or the baby is born with stunting and both mental and physical disabilities [10, 11]. Other conditions are such as abortion, premature delivery, pre-eclampsia, placental abruption, and lack of concern for the foetus [12–15]. Not only does it have an impact on the growth and development of the foetus, but also this condition has an impact on the condition of marital satisfaction, which leads to miscommunication and disharmony [16, 17]. The previous study mentioned a relationship between the foetus and the transition to parenthood during pregnancy [18, 19]. The exceptional relationships between pregnant women, their husbands, and foetuses are placed at the core of the living environment that needs to be strengthened to maintain their wellness [20, 21].

The couple's orientation to everyday life, as well as changes to daily life, joyfulness experienced, and their subsequent adaptation to those changes, is reflected in their marital satisfaction [22]. Couples who can talk and work out their differences on key marital issues to the satisfaction of both partners have a harmonious marriage [23, 24]. During pregnancy, marital satisfaction plays an important role in maternal psychology, foetal development, and also the coverage of vaccination for baby well-being [25–28]. To reach this, husbands should understand the emotional condition of pregnant women caused by hormonal fluctuations [29]. Becoming a mother theory from Mercer [30] states that, at the stages of commitment, attachment and preparation, it can increase the bond between mother and children as well as environmental factors that also have an impact on husband-wife marital satisfaction.

To increase the closeness of the relationship between parents and foetus as well as to increase marital harmony, mindfulness interventions can be used as a reference. These interventions can reduce psychological distress and improve emotion regulation and empathy [31]. Previous study mentioned high correlation between mindfulness and marital satisfaction [32]. Other studies about mindfulness that that specific on childbirth and parenting have a positive improvement in mother's wellbeing and parents' relationship [33, 34]. Mindful technique helps both husband and wife to acknowledge any perceptions, problems, and recognise any disadvantages and advantages in their marital life. This technique also helps to identify the positive perception and satisfying marital relationship by creating open-minded discussion to solve any problems, including pregnancy condition [35, 36].

To date, research indicates that the utilisation of mobile health (m-health) in providing nursing interventions for pregnant women is possible [37]. Technically, mindfulness intervention that emphasises on maternal relationship is implemented manually and online [38–40]. Thus, we developed mHealth-based mindfulness namely "Fostering the Emotional Bond of Parents and Foetus/*Menjalin Ikatan Emosional Orangtua dan Janin* (MIESRA)" to facilitate pregnant women and husbands that can be implemented everywhere and anytime to maintain the marital life during pregnancy for mothers' and foetus wellbeing. MIESRA mHealth provides many interventions to help the mother maintain the pregnancy and foetal condition and marital satisfaction for both husband and wife. This mHealth can be used by both husband and wife. Thus, we conducted this study with aim to assess the effect of MIESRA as a mobile health-based nursing intervention to increase the satisfaction in a husband-wife relationship.

## Materials and methods

### Study design

We utilised a quasi-experimental study to evaluate the effectiveness of MIESRA mHealth to the outcome variable [41, 42]. This study design is appropriate since the author did not perform randomisation. We faced limitation on study settings and subjects' recruitment during the COVID-19 pandemic in Indonesia.

### Setting and sample

The study is conducted in the government and private hospitals in Jakarta, Indonesia. These hospitals were elected as a referral mother and children hospital in Indonesia. For homogeneity reasons, the data collection for all groups was implemented in both hospitals (Fig 1). We performed the study from baseline to three months (May to July 2021).

A total of 93 couples (husband and wife) participated in this study. A purposive sampling technique was used to recruit the participants. A total sample was estimated of 10% dropping out from this study. We used G*Power (software used to calculate statistical power) version 3.1.9.6 with the assumption of $p < 0.05$, 0.9 of power estimation, and 0.8 of effect size [43, 44]. Furthermore, we divided into three groups, namely couple groups, single groups, and control groups. Each group consisted of 31 couples. The inclusion criteria consisted of 1) pregnant women with *primigravida* or *multipara*; 2) a minimum of secondary education level; 3)

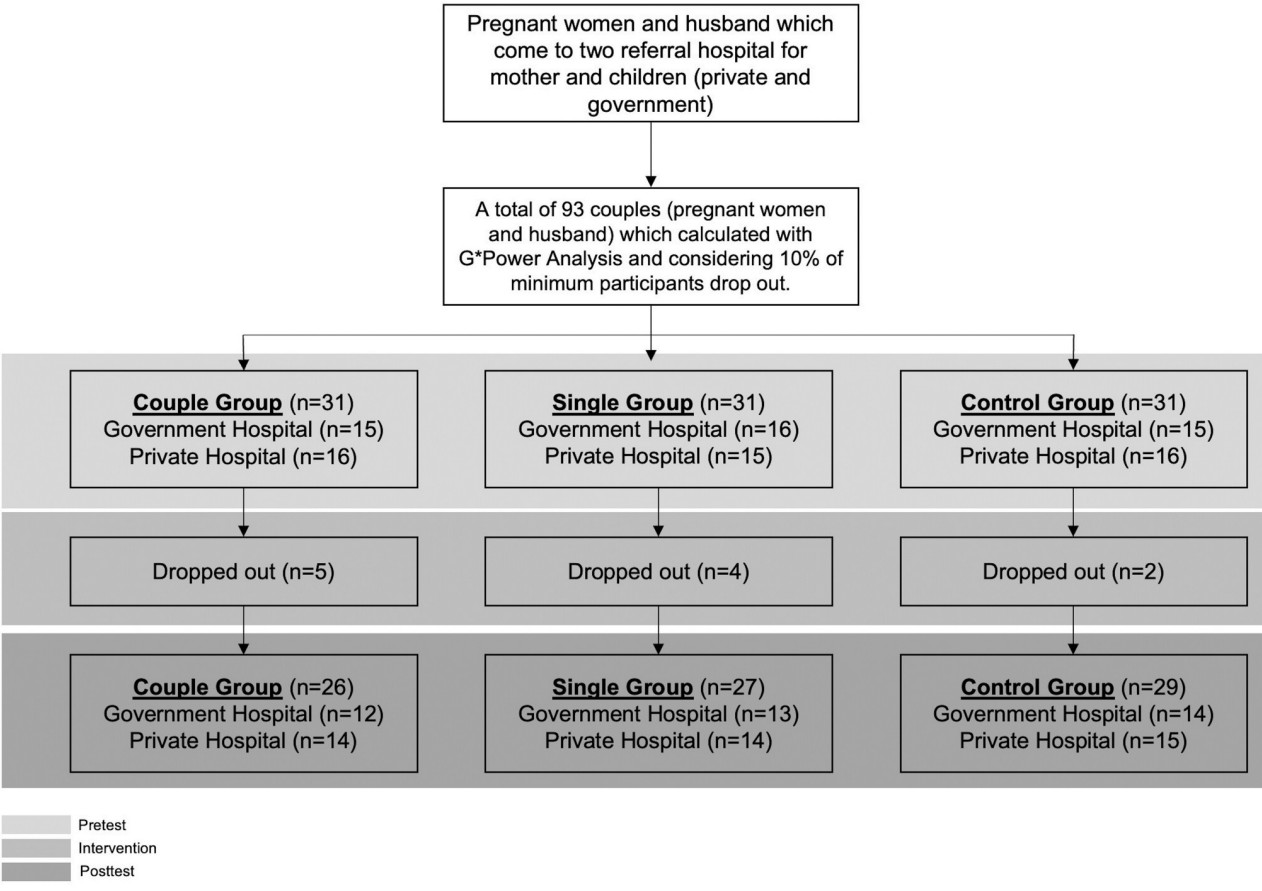

**Fig 1. Sample diagram flow chart.**

gestational age between 20 and 34 weeks; 4) pregnant women aged 20–45 years; 5) husband willing to participate; 6) couple has smartphone with Android-based system; 7) never diagnosed with or under medication for mental health problems. In addition, couples who drop out during the study process will be excluded. At the end, 11 couples e dropped out (two couples resigned during the study process and nine mothers were delivered before the post-test data were collected). Finally, a total of 82 couples was included (Fig 1).

## Instruments

We used Compatibility of Husband-and-Wife Relationships / *Kesesuaian Hubungan Suami Istri* (KHSI) questionnaire to observe marital satisfaction [45]. This questionnaire was used by the author as a consideration with participants' characteristics in Indonesia. This instrument consists of 29 items that cover six components 1) closeness; 2) conformity; 3) understanding; 4) a reflection of affection; 5) satisfaction; and 6) a husband-wife togetherness. In favourable statements, the score starts from 0 to 3 (items no. 1–14, 16,19, 20, 23–29). In the negative statements, the score starts from 3 to 0 (items no. 15, 17, 18, 21, 22). A lower score indicated satisfied and high score indicated not satisfied. A mean cut-off point was used to interpret the score of KHSI. The value of validity and the internal confidence consistency had a Cronbach's alpha of 0.86. A reliability test of KHSI was conducted on 116 pregnant women in urban areas with Cronbach's alpha of 0.89 [46].

**MIESRA mHealth.** MIESRA mHealth is an interactive learning media for pregnant women and their husbands that can guide how to build emotional bonds between parents and foetuses. This application is built using Android Studio version 4.1.2 which can be installed on devices such as mobile phones with Android Operating System 4.0 or the latest. This application consists of two types, namely applications for wives and applications for husbands and can be downloaded from https://miesra-app.nfcworld.web.id/. Detailed information can be seen in Table 1 & S1 Data. MIESRA mHealth has been tested through several steps. 1) At the beginning, MIESRA was tested to five users to evaluate the menus and function in detail. 2) The second steps, a limited test to five couples (pregnant women and husband), was used to evaluate the applicability and effectivity of application to reach the purpose. 3) The next step is the evaluation on efficacy, efficiency, and usability using Heuristic Evaluation (HE) approach [47]. HE was performed by an expert in nursing informatics and working as an analyst in the Digital Transformation Office at the Ministry of Health of Republic of Indonesia. Based on the HE assessment, MIESRA mHealth received 0.6 value (1 is the highest value) which means that MIESRA needed a minor revision and can be approved after revision. 5) In the last steps, a

**Table 1. MIESRA mHealth content information.**

| Indicator | Detail | Wife | Husband |
|---|---|---|---|
| Apps instruction | Contain a schematic of the steps to be carried out | ✓ | ✓ |
| Pretest and posttest | Contain three components of an assessment of the condition of pregnant women including psychological health, emotional bonding of mother and foetus and satisfaction of husband-wife relationship. | ✓ | ✓ |
| Daily activities | Consist of three components. 1) The video contains how to build an emotional bond between parents and their foetus through Mindfulness techniques, namely the technique of practicing thoughts, feelings and being aware of the condition of the pregnancy through interaction with the foetus, which is guided by a facilitator using audio media and a video dissertation on how to do it; 2) Mindfulness audio; and 3) Mother's dairy notes. | ✓ | ✗ |
| Education | Contains health education materials on how to form emotional bonds consisting of 1) Pregnancy period (fetal development, physical changes in pregnant women, psychological changes, and how to bond during pregnancy between couples and with the foetus); 2) Period after childbirth (mother's physical changes after childbirth, psychological changes, and how to bond during pregnancy between couples and with the baby). | ✓ | ✓ |
| Documentation | Consist of information about the result of pre and posttest value and suggestion to improve the health condition. | ✓ | ✓ |

trial test used System Usability Scale (SUS) [48] to 15 couples in the government hospitals. Based on SUS test, MIESRA mHealth received 80.1 score, which means MIESRA is acceptable and can be used with the excellence category. We obtained Intellectual Property Rights approval from the Indonesian Ministry of Law and Human Rights (No. 000291880).

## Study procedures

During the COVID-19 pandemic, we performed the study in two referral hospital in Jakarta, Indonesia. We set the study procedure with the condition of COVID-19 by following the Indonesia Government and hospital rules. We conducted the assessment from baseline to three months. At the beginning, we explained all the information about the study purpose and procedure to the respective respondents and asked them manually to sign the informed consent. Then, all the respondents in the couple and single groups were asked to download the MIESRA mHealth to their own mobile phone. All participants in the three groups received the same intervention in the form of prenatal care, education, and consultation from an obstetrician (See Fig 2). Respondents were allowed to consult with researchers via telephone or chat.

The pre-test was conducted to find out initial information regarding marital satisfaction through the MIESRA mHealth, except for the control group who used a special website provided by the researcher. Researchers could monitor all developments and information on the MIESRA mHealth through a special account.

In the couple group, the wife did mindfulness based on the information in the MIESRA mHealth together with her husband. This intervention was carried out based on the mindfulness guide available in the wife-MISRA mHealth. Husband and wife simultaneously viewed the video as an initial guide to mindfulness. Then, they were asked to listen to mindfulness-related audio from the wife-MIESRA mHealth and carry out the instructions with their husbands. This intervention was two times a week for four weeks with 15 minutes of each session. In addition, there is a Diary feature in the wife-MIESRA mHealth that helps the wife to write, submit complaints, and progress during the pregnancy process. This feature can be viewed

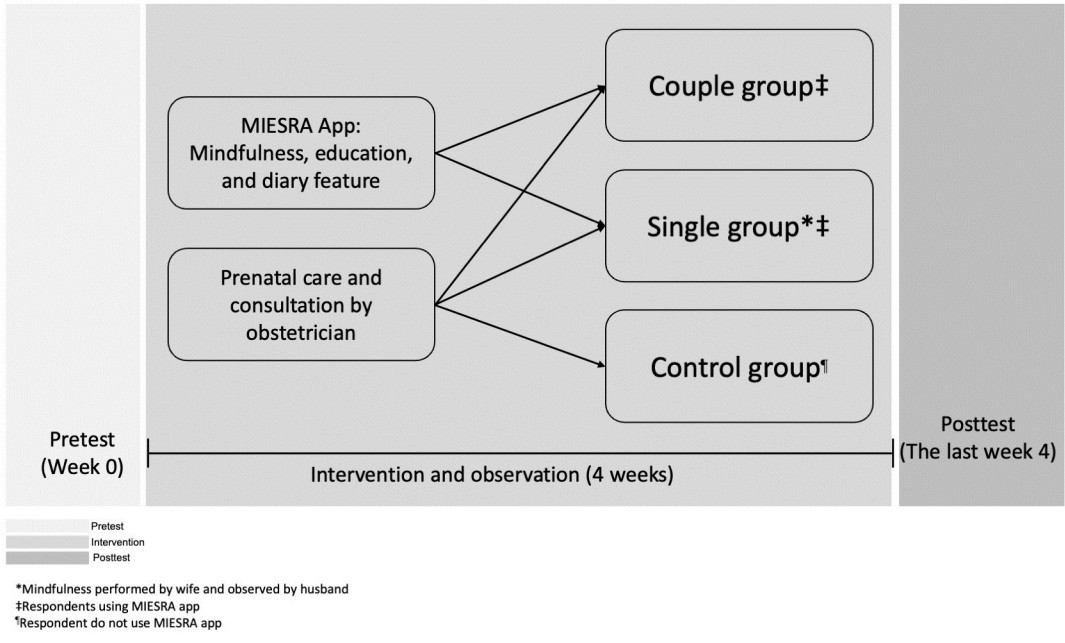

**Fig 2. Study procedure diagram flow.**

and accessed by husbands (using the husband-MIESRA mHealth) with the aim to increase the husband's knowledge regarding their wife's condition during the pregnancy. Researchers monitor the activities carried out by couples through the application media and will remind them through the media in the app or by telephone if they do not intervene. Then, post-test was performed after four weeks of intervention.

In a single group, the wife sees the video as an initial guide to doing mindfulness. Then they listen to mindfulness-related audio and do it according to the instructions. In this group, the husband did not participate in mindfulness and only monitored his wife. This intervention was two times a week for four weeks with 15 minutes of each session. Then, the wife filled out a diary on the MIESRA mHealth feature to convey what she feels during pregnancy. In this group, husbands received education through the husband-MIESRA mHealth and could directly monitor the contents of the wife's diary using the husband-MIESRA mHealth. Then post-test was done after four weeks of intervention.

In the control group, respondents received programme interventions from hospitals which included education and consultation with obstetricians. What makes this group different is that they didn't get the MIESRA mHealth facility. Couples were allowed to ask researchers related to the research process via telephone or chat. Next, the post-test was conducted after four weeks and respondents received the MIESRA mHealth facility and an explanation of how to use it from the researcher.

## Data analysis

IBM SPSS version 24 was used to conduct statistical analysis. The descriptive characteristics of the respondents in all groups were represented using descriptive statistics such as numbers, percentages, and homogeneity test. Thus, one-way ANOVA test was utilised to determine the likelihood of homogeneity in all groups (homogeneity $p = >0.05$). Furthermore, we performed Generalized Estimating Equations (GEE) of multivariate analysis to evaluate the effectiveness of MIESRA mHealth. GEE is a development of the Generalized Linear Model (GLM) which is used to measure data that contain autocorrelation. The dependent variable can be in the form of numeric or categorical data and can control covariate variables that are dynamic or changing, whereas the GLM can only measure the numerical dependent variable and the data are static [49, 50]. In this study, the covariate contained a dynamic dependent variable because the pre-test value was different from the characteristic value, which was static. During the statistical analysis, we performed and consulted to the statistical expert in Universitas Indonesia.

## Ethical consideration

This research was ethically approved by the Human Research Ethics Committee of the Nursing Faculty, Universitas Indonesia (No. SK-243/UN2.F12.D1.2.1/ETIK. FIK. 2019) and the hospitals' ethical committee. All respondents were over 18 years old, their privacy protected, and voluntarily authorised the written informed consent. The individual in this study has given written informed consent to publish this case detail. The individuals pictured in S1 Data have provided written informed consent (as outlined in the PLOS consent form) to publish their images alongside the manuscript, and the animated images presented in the S1 Data were drawn by the author.

## Results

From the total 82 couples included in this study, we present the demographic data in women and husband separately. We found that women are predominantly in the low risk aged from 20 to 35 years. Most of the women in the three groups completed college or university and

have planned pregnancies. An almost similar number of women works outside home as with the homemaker ones. More women in the intervention groups live with their partners and children only rather than with extended family, but these are different with the control group. Overall, the characteristics among women in the groups shows homogeneity (p-value >.05), unless for the parity. In the husband category, the majority are young adults (age 26 to 35 years), had university education level, work as a private employee, and earn above the Jakarta's regional minimum wage of IDR 4,200,000). Statistically, there are no differences in the characteristics of the father among the groups (p-value >0.05). Thus, the three groups in this study are homogenic (Table 2).

The homogeneity test of harmonious relationship score of the female and male groups before the intervention reported that the three groups (single, paired, and control) showed similar KHSI scores pre-intervention (p-value = 0.707 for women and p = 0.394 for men). Therefore, the differences that occurred post-intervention were influenced by the adoption of mindfulness exercises through MIESRA mHealth (Table 3).

It can be seen in Table 4 that there were two variables included in the women's KHSI multivariate test (intervention variables and husband's KHSI). Three variables were included in the husband's multivariate test (intervention variables, husband's age, and women's KHSI).

### MIESRA mHealth towards women's satisfaction score

Both the complete model and the final model of the variables that affect the women's KHSI score after the intervention were the provision of the intervention, time, and the husband's KHSI with a significance value lower than 0.001. The final model showed the women's KHSI score was 3.68 points better than before the intervention. The husband's KHSI also affects the mother's KHSI condition. After the intervention, the mother's KHSI score in the paired intervention group was 7.46 points better than the control group. In the individual intervention group, the mother's KHSI condition was 9.11 points better than the control group after the intervention (Table 5).

### MIESRA mHealth towards husband's satisfaction score

Both the complete model and the final model of the variables affecting the KHSI score after the intervention are the provision of the intervention, time and the women's KHSI with a value of significantly less than 0.001. The final model showed that the husband's KHSI score was 2.29 points better than before the intervention. The mother's KHSI also affects the husband's KHSI condition. After the intervention, the husband's KHSI score in the paired intervention group was 7.04 points better than the control group. In the individual group, after the intervention, the husband's KHSI condition was 3.74 points better than the control group (Table 6).

## Discussion

Marital satisfaction during pregnancy is needed as a husband's support system. This has an impact on the psychology of the mother and the development of the foetus. For this reason, the MIESRA mHealth was created to help increase marital satisfaction among couples during pregnancy by using a mindfulness approach. Pregnant women used the MIESRA mHealth to establish emotional bonds between parents and foetuses with partners or individually.

In our study, we found that respondents in the paired and single group showed a better result in KHSI score compared to control group. It means that the MIESRA mHealth has a positive effect on marital satisfaction. As we mentioned that the MIESRA mHealth was developed based on mindfulness approach, the finding is supported by previous study using Mindfulness-Based Childbirth and Parenting (MBCP) programme that also increases the marital

**Table 2. Demographic and pregnancy characteristics of the respondents (n = 82 couples).**

| Demographic Characteristics | Control group (n = 29 couples) | Single group (n = 27 couples) | Paired group (n = 26 couples) | Homogeneity Test |
|---|---|---|---|---|
| | n (%) | n (%) | n (%) | p-value |
| Pregnant Women: | | | | |
| Age | | | | |
| • Low risk (20–35 years) | 25 (86) | 24 (89) | 23 (88) | (0.90) |
| • High risk (<20 or >35 years) | 4 (14) | 3 (11) | 3 (12) | |
| Last education | | | | |
| • High school | 8 (38) | 7 (26) | 8 (31) | (0.85) |
| • University/college | 21 (72) | 20 (74) | 18 (69) | |
| Occupation | | | | |
| • Homemaker | 15 (52) | 14 (52) | 12 (46) | (0.80) |
| • Working outside home | 14 (48) | 13 (48) | 14 (54) | |
| Parity | | | | |
| • Primipara | 16 (55) | 11 (41) | 18 (69) | (0.01) |
| • Multipara | 13 (45) | 16 (59) | 8 (31) | |
| Pregnancy | | | | |
| • Planned | 21 (72) | 14 (52) | 15 (58) | (0.07) |
| • Unplanned | 8 (28) | 13 (48) | 11 (42) | |
| Type of family | | | | |
| • Nuclear | 14 (48) | 16 (59) | 14 (54) | (0.51) |
| • Extended | 15 (52) | 11 (41) | 12 (46) | |
| Husband: | | | | |
| Age | | | | |
| • Youths (17–25 year | 2 (7) | 1 (4) | 3 (12) | (0.69) |
| • Young adults (26–35 year) | 19 (65) | 21 (78) | 16 (62) | |
| • Middle-aged adults (36–45 year) | 8 (28) | 5 (19) | 7 (27) | |
| Last education | | | | |
| • High School | 7 (24) | 6 (22) | 9 (35) | (0.55) |
| • University | 22 (76) | 21 (78) | 17 (65) | |
| Occupation | | | | |
| • Civil servants | 3 (10) | 5 (18) | 5 (19) | (0.62) |
| • Private employee | 19 (66) | 20 (74) | 17 (65) | |
| • Entrepreneur | 4 (14) | 2 (8) | 2 (8) | |
| • Labor | 3 (10) | 0 | 2 (8) | |
| Income | | | | |
| • High (< IDR 4,200,000) | 22 (76) | 22 (81) | 18 (69) | (0.64) |
| • Low (≥ IDR 4,200,000) | 7 (24) | 5 (19) | 8 (31) | |

satisfaction [33]. MIESRA mHealth increased communication and mutual understanding of partners as experienced during childbirth. Husband was fully present in helping from preparation for delivery to completion of labour by reminding and guiding the wife in practising mindfulness activities. This is similar to previous study which mentioned that mindfulness practice during pregnancy increased the interpersonal relationship and physical effect, such as relaxation and comfort with the husband [51].

**Table 3. Husband and wife satisfaction score before and after MIESRA mHealth intervention (n = 82).**

| Variable | Control group (n = 29) | | | Individual Intervention (n = 27) | | | Paired Intervention (n = 26) | | | Homogeneity Pretest: F-score (p-value) |
|---|---|---|---|---|---|---|---|---|---|---|
| | Pretest | Posttest | Mean difference | Pretest | Posttest | Mean difference | Pretest | Posttest | Mean difference | |
| **Women Score** | | | | | | | | | | |
| Mean | 12.79 | 18.41 | 5.62 | 15.04 | 7.81 | -7.23 | 15.38 | 8.27 | -7.11 | 0.35 |
| SD | 8.21 | 10.91 | (<0.001) | 10.18 | 5.71 | (<0.001) | 11.12 | 5.1 | (<0.001) | -0.707 |
| **Husband's Score** | | | | | | | | | | |
| Mean | 11.76 | 15.86 | 4.11 | 13.11 | 9.33 | -3.78 | 13.35 | 6.308 | -7.04 | 0.94 |
| SD | 1.61 | 1.3 | -0.003 | 1.67 | 1.34 | -0.006 | 1.7 | 1.37 | -0.005 | -0.394 |

**Table 4. Candidate confounding variables in modeling the effects of MIESRA mHealth on satisfaction of husband-wife relationships.**

| | Women's KHSI score | | Husband's KHSI score |
|---|---|---|---|
| **Static Independent Variables** | **p-value** | **Static Independent Variables** | **p-value** |
| Intervention | 0.001* | Intervention | <0.001* |
| Age | 0.818 | Age | 0.001* |
| Education | 0.483 | Education | 0.707 |
| Occupation | 0.818 | Occupation | 0.661 |
| Parity | 0.522 | Income | 0.633 |
| Pregnancy plan | 0.775 | | |
| Family living together | 0.671 | | |
| Abortion history | 0.506 | | |
| Pregnancy problem | 0.812 | | |
| **Dynamic Independent Variables** | | | |
| Husband's KHSI | <0.001* | Women's KHSI | <0.001* |

*Significance <0.01

**Table 5. Modeling effect of MIESRA mHealth on women's satisfaction score.**

| Independent Variable | Full | Model | p | Final | Model | p |
|---|---|---|---|---|---|---|
| | Coef. B | 95% CI | | Coef. B | 95% CI | |
| Intercept | 7.13 | | 0.004 | 7.23 | | <0.001 |
| Intervention | | | | | | |
| Paired | 1.83 | -2.70–6.36 | 0.427 | 1.84 | -2.67–6.35 | 0.424 |
| Individual | 1.59 | -2.61–5.78 | 0.459 | 1.6 | -2.71–5.92 | 0.467 |
| Control | Ref. | | | Ref. | | |
| Measurement Time | | | | | | |
| Posttest | -3.65 | 0.75–6.54 | 0.014 | 3.68 | 0.99–6.36 | 0.007 |
| Pretest | Ref. | | | Ref. | | |
| Husband's KHSI | 0.47 | 0.22–0.72 | <0.001 | 0.47 | 0.22–0.73 | <0.001 |
| Intervention*Time | | | | | | |
| Paired*Posttest | -7.34 | -12.89 –(-1.86) | 0.009 | -7.46 | -12.10 –(-2.83) | 0.002 |
| Individual*Posttest | -9.05 | -13.33 –(-4.77) | <0.001 | -9.11 | -13.25 –(-4.98) | 0.001 |
| Control*Posttest | Ref. | | | Ref. | | |

0.0 = Reference

**Table 6. Modeling effect of MIESRA mHealth on husband's satisfaction score.**

| Independent Variable | Full | Model | p | Final | Model | p |
|---|---|---|---|---|---|---|
| | Coef. B | 95% CI | | Coef. B | 95% CI | |
| Intercept | 10.41 | | 0.044 | 7.64 | | <0.001 |
| Intervention | | | | | | |
| Paired | 1.2 | -2.79–5.19 | 0.555 | 1.75 | -3.07–4.57 | 0.699 |
| Individual | 0.87 | -2.90–4.63 | 0.652 | 0.63 | -3.24–4.50 | 0.75 |
| Control | Ref. | | | Ref. | | |
| Measurement Time | | | | | | |
| Posttest | 1.98 | 0.53–3.42 | 0.007 | 2.29 | 0.86–3.73 | 0.002 |
| Pretest | Ref. | | | Ref. | | |
| Women's KHSI | 0.29 | 0.05–0.53 | 0.018 | 0.32 | 0.07–0.58 | 0.012 |
| Intervention*time | | | | | | |
| Paired*Posttest | -5.84 | -9.16 –(-2.52) | <0.001 | -7.04 | -9.75 –(-4.32) | <0.001 |
| Individual*Posttest | -2.29 | -6.04–1.47 | 0.232 | -3.74 | -6.99 –(-0.50) | 0.024 |
| Control*Posttest | Ref. | | | Ref. | | |

The success of mindfulness practice in increasing the marital relationship does not have to be performed manually in the class. It can be assisted by mHealth as described in this current study. A previous study mentioned online mindfulness as new methods to be more applicable in an easy way [40]. Moreover, mindfulness practice resulted in quality marital satisfaction and psychological wellbeing [38, 39, 52]. Mindfulness exercises carried out in the early stages have shown significant results in improving the quality of husband-and-wife relationships [53]. Mindfulness can effect to marital satisfaction through psychoeducation, daily registration and assessment of daily negative experiences, and a short mindfulness practice (5–15 minutes per day) [35, 38]. Those steps are important in mindfulness to increase the understanding from both couple and lead to happy marital satisfaction and wellbeing in general.

The improvement in the father's KHSI score in the paired intervention group is better than in the single group. Our finding shows that mindfulness practice interventions carried out jointly by married couples produce a better result than individual interventions. Therefore, we suggested that pregnant women prioritise mindfulness exercises with their husbands to achieve a maximum harmonious relationship. Previous studies mention that mindfulness performed with couples leads to effective communication, understanding, and support of each other [54, 55]. Another study also mentioned that it can increase attention, affection, and wellbeing and improve quality of life [56]. Likewise, mindfulness exercises given to pregnant women with their husbands significantly affect their ability to act consciously (acting with awareness) and increase satisfaction and quality of relationships with partners [55]. During the study, husbands were more intense in participating in the training as they directly obtained information from health providers. Transparent information from these professionals makes fathers understand the right way and feel a deeper relationship with their wives and children and a closer relationship with the foetus [57].

Mindfulness for marital harmony has a very positive effect for mother, foetus, and husband. The mother showed behavioural changes, which are more patient and calmer in interacting and communicating with partners. The husband described that they were more concerned and alert to providing support, assistance, and involvement in the process of pregnancy and childbirth for their wives [51]. Meanwhile, the foetus also received a positive impact on growth and development [58]. In the maternity nursing area, mindfulness exercises for men improving interactions between father, mother, and foetus can further enhance harmonious

relationships. Interactions are conducted by being alert to every moment of activity [57]. The father's ability to feel the presence of the foetus can significantly increase the closeness of the emotional relationship between the father and the foetus [59], raise the man's involvement in carrying out his role as a father [60], and increase the harmonious relationship of the father with a partner [61].

In a mindful state, the mind, emotions, and body sensations look at others with open and non-judgemental thought [61]. The quality benefits increase relationship attention, making the relationship healthier, reducing negative interaction, and increasing positive environment for the pregnancy [62]. Mindfulness nursing interventions that are performed individually affect partners to be more responsive because the principle of being fully present and non-judgemental causes a good perception of partner behaviour [40, 63]. This condition was also experienced by the husbands of the individual group in this current study. Even though these men did not get mindfulness training like the husbands of the paired group, they still had an impact on improving the value of the harmony of the husband-and-wife relationship that the father obtained.

## Strengths and limitations

The strengths of this study include (1) the intervention utilizing mHealth media, which allows respondents to access it without limitations of time and place (2) MIESRA mHealth, an application that provides crucial information during pregnancy, pre-test and posttest assessments for knowledge, and daily activities comprising instructional videos on mindfulness, audio materials, and diary notes, (3) MIESRA mHealth is easy to install, use, and comprehend, (4) detailed research methods were employed, and (5) statistical tests conducted by experts to ensure robust results. Additionally, the availability of this application can support the limited number and time constraints of health workers in providing antenatal health services. It enables monitoring of the psychological health of pregnant women and their partners, thereby enhancing the emotional bond between the mother and the foetus.

However, this study had several limitations, including (1) the participants were limited to a specific area. Therefore, conducting research in broader locations while considering beliefs and culture becomes essential; (2) MIESRA mHealth still had certain limitations. Researchers were unable to examine it in real-time to determine when participants engaged in mindfulness interventions and learning activities such as reading educational materials or watching educational videos. This aspect needs improvement in future work; (3) The interventions in this study were a single package; therefore, specific interventions were not conducted each week; (4) The generalization of research results is limited to pregnant women and partners without mental health disruptions. Further research is recommended for groups of patients diagnosed with mental illnesses. Overall, these limitations highlight the areas for improvement and suggest potential directions for future study.

## Conclusion

Nursing interventions to build emotional bonds between parents and foetuses based on mHealth can be a promising intervention for marital harmony during the perinatal period. The husband's active involvement in the mindfulness exercise programme with his wife and interacting with the foetus yields psychological comfort and increases emotional bonds with his wife and foetus. Hence, the MIESRA mHealth can be one of solutions and suggestions to the Indonesia Government especially to increase the contribution of husbands in prenatal, antenatal, and postnatal care. We recommend the MIESRA mHealth as a nursing intervention that can implemented during pregnancy is integrated into the antenatal care programme to increase the marital satisfaction, foetal growth, and wellbeing in general.

## Supporting information

**S1 Data. MIESRA mHealth information and description.**
(PDF)

**S2 Data.**
(XLSX)

**S1 File.**
(DOCX)

## Acknowledgments

We acknowledge to all respondents who contributed to the study and Universitas Indonesia, and Hospital who supported the study.

## Author Contributions

**Conceptualization:** Besral Besral, Misrawati Misrawati, Yati Afiyanti.

**Data curation:** Misrawati Misrawati, Raden Irawati Ismail, Hidayat Arifin.

**Formal analysis:** Besral Besral, Misrawati Misrawati, Yati Afiyanti.

**Funding acquisition:** Yati Afiyanti.

**Investigation:** Besral Besral, Misrawati Misrawati.

**Methodology:** Besral Besral, Misrawati Misrawati, Yati Afiyanti.

**Resources:** Raden Irawati Ismail, Hidayat Arifin.

**Software:** Misrawati Misrawati.

**Validation:** Raden Irawati Ismail, Hidayat Arifin.

**Visualization:** Raden Irawati Ismail, Hidayat Arifin.

**Writing – original draft:** Besral Besral, Misrawati Misrawati, Yati Afiyanti, Raden Irawati Ismail, Hidayat Arifin.

**Writing – review & editing:** Besral Besral, Misrawati Misrawati, Yati Afiyanti, Raden Irawati Ismail, Hidayat Arifin.

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
