## [Decision Letter · Decision Letter 0]

28 Apr 2023

PONE-D-23-01469MIESRA mHealth: Marital Satisfaction During PregnancyPLOS ONE

Dear Dr. Misrawati,

Thank you for submitting your manuscript to PLOS ONE. After careful consideration, we feel that it has merit but does not fully meet PLOS ONE’s publication criteria as it currently stands. Therefore, we invite you to submit a revised version of the manuscript that addresses the points raised during the review process.

We look forward to receiving your revised manuscript.

Kind regards,

Giulia Ballarotto

Academic Editor

PLOS ONE

Journal Requirements:

3. Please include a caption for figure 2.

4. We note that Figure S1 and S2 includes an image of a patient/participant in the study. 

Reviewers' comments:

Reviewer's Responses to Questions

**Comments to the Author**

1. Is the manuscript technically sound, and do the data support the conclusions?

Reviewer #1: Yes

Reviewer #2: Yes

2. Has the statistical analysis been performed appropriately and rigorously? 

Reviewer #1: Yes

Reviewer #2: Yes

3. Have the authors made all data underlying the findings in their manuscript fully available?

Reviewer #1: Yes

Reviewer #2: Yes

4. Is the manuscript presented in an intelligible fashion and written in standard English?

Reviewer #1: Yes

Reviewer #2: Yes

5. Review Comments to the Author

Reviewer #1: Very interesting and good topic that had been discussed. The importance of the study had been highlighted clearly at the begining of the writing. How the implementation of the treatments for all three types of group had been discussed. The inclusion criteria for the sample also clearly stated. However, would like to know is the information of income being asked to the pregnant women? If no, could the author provide justification. Then, the experiment being done for 4 weeks, what would be the details of the content for each week? Would be good if can add it in writing. Is it standardized with the other group regarding the content exposed for each of the weeks? Lastly suggest to highlight the strength of the MIESRA in the strenght and limitation.

Reviewer #2: Thank you for the opportunity to review this study. I think the topic is very important, as has been well pointed out by the authors. Overall, the paper was well written and there are only a few points that could be revised.

Specifically, in reviewing the existing literature, it would be useful to look more into the transition that women go through. I cite some studies as examples:

Tambelli, R., Ballarotto, G., Trumello, C., & Babore, A. (2022). Transition to Motherhood: A Study on the Association between Somatic Symptoms during Pregnancy and Post-Partum Anxiety and Depression Symptoms. International Journal of Environmental Research and Public Health, 19(19), 12861.

Hennekam, S., Syed, J., Ali, F., & Dumazert, J. P. (2019). A multilevel perspective of the identity transition to motherhood. Gender, Work & Organization, 26(7), 915-933.

Trentini, C., Pagani, M., Lauriola, M., & Tambelli, R. (2020). Neural responses to infant emotions and emotional self-awareness in mothers and fathers during pregnancy. International journal of environmental research and public health, 17(9), 3314.

Ammaniti, M., Trentini, C., Menozzi, F., & Tambelli, R. (2014). Transition to parenthood: Studies of intersubjectivity in mothers and fathers.

Duarte-Guterman, P., Leuner, B., & Galea, L. A. (2019). The long and short term effects of motherhood on the brain. Frontiers in neuroendocrinology, 53, 100740.

In addition, it would be helpful if the authors provided more information regarding the content of the application.

Finally, the authors are invited to highlight strengths and weaknesses of MIESRA, but especially the future directions that the study's findings imply

6. PLOS authors have the option to publish the peer review history of their article (what does this mean?). If published, this will include your full peer review and any attached files.

Reviewer #1: No

Reviewer #2: No

---

## [Author Response · Author response to Decision Letter 0]

25 May 2023

Response to Editor and Reviewers

To Editor

Response: Thank you, we have followed the guidelines.

2. In your Data Availability statement, you have not specified where the minimal data set underlying the results described in your manuscript can be found. PLOS defines a study's minimal data set as the underlying data used to reach the conclusions drawn in the manuscript and any additional data required to replicate the reported study findings in their entirety. 

Response: Thank you, we have uploaded the raw data of this study. 

“All data relevant to the study are included in the article and its Supporting Information files”

3. Please include a caption for figure 2

Response: Thank you, we have added the caption for figure 2. 

4. We note that Figure S1 and S2 includes an image of a patient/participant in the study. 

Response: Thank you for the correction. We have filled the consent form for publication and mentioned the statement in the ethical consideration's part. 

Response: Thank you, we have added the supporting information.

6. Please review your reference list to ensure that it is complete and correct

Response: Revised. Thank you,

Reviewer 1

1. Very interesting and good topic that had been discussed. The importance of the study had been highlighted clearly at the begining of the writing. 

Response: Thank you, 

2. How the implementation of the treatments for all three types of group had been discussed. The inclusion criteria for the sample also clearly stated. However, would like to know is the information of income being asked to the pregnant women? If no, could the author provide justification. 

Response: Thank you for your question. We did not ask pregnant women about their income because, for our participants, the primary source of income was their husbands. Therefore, we refrain from inquiring about income from pregnant women. 

3. Then, the experiment being done for 4 weeks, what would be the details of the content for each week? Would be good if can add it in writing. 

Response: Thank you for the question. In this study, we are unable to provide detailed information about the intervention content for each week because the interventions in our study were implemented as a single package. We have included the details of the study procedure in the study procedure section. However, this is a valuable suggestion, so we have added this information to the study limitations. 

4. Is it standardized with the other group regarding the content exposed for each of the weeks? 

Response: Yes, the contents were standardized. 

5. Lastly suggest to highlight the strength of the MIESRA in the strenght and limitation.

Response: Thank you for the suggestion. We have revised the strengths and limitation of this study. 

“The strengths of this study include (1) the intervention utilizing mHealth media, which allows respondents to access it without limitations of time and place (2) MIESRA mHealth, an application that provides crucial information during pregnancy, pre-test and posttest assessments for knowledge, and daily activities comprising instructional videos on mindfulness, audio materials, and diary notes, (3) MIESRA mHealth is easy to install, use, and comprehend, (4) detailed research methods were employed, and (5) statistical tests conducted by experts to ensure robust results. Additionally, the availability of this application can support the limited number and time constraints of health workers in providing antenatal health services. It enables monitoring of the psychological health of pregnant women and their partners, thereby enhancing the emotional bond between the mother and the foetus.

 However, this study had several limitations, including (1) the participants were limited to a specific area. Therefore, conducting research in broader locations while considering beliefs and culture becomes essential; (2) MIESRA mHealth still had certain limitations. Researchers were unable to examine it in real-time to determine when participants engaged in mindfulness interventions and learning activities such as reading educational materials or watching educational videos. This aspect needs improvement in future work; (3) The interventions in this study were a single package; therefore, specific interventions were not conducted each week; (4) The generalization of research results is limited to pregnant women and partners without mental health disruptions. Further research is recommended for groups of patients diagnosed with mental illnesses. Overall, these limitations highlight the areas for improvement and suggest potential directions for future study.”

Reviewer 2

1. Thank you for the opportunity to review this study. I think the topic is very important, as has been well pointed out by the authors. Overall, the paper was well written and there are only a few points that could be revised.

Response: Thank you,

2. Specifically, in reviewing the existing literature, it would be useful to look more into the transition that women go through. I cite some studies as examples:

• Tambelli, R., Ballarotto, G., Trumello, C., & Babore, A. (2022). Transition to Motherhood: A Study on the Association between Somatic Symptoms during Pregnancy and Post-Partum Anxiety and Depression Symptoms. International Journal of Environmental Research and Public Health, 19(19), 12861.

• Hennekam, S., Syed, J., Ali, F., & Dumazert, J. P. (2019). A multilevel perspective of the identity transition to motherhood. Gender, Work & Organization, 26(7), 915-933.

• Trentini, C., Pagani, M., Lauriola, M., & Tambelli, R. (2020). Neural responses to infant emotions and emotional self-awareness in mothers and fathers during pregnancy. International journal of environmental research and public health, 17(9), 3314.

• Ammaniti, M., Trentini, C., Menozzi, F., & Tambelli, R. (2014). Transition to parenthood: Studies of intersubjectivity in mothers and fathers.

• Duarte-Guterman, P., Leuner, B., & Galea, L. A. (2019). The long and short term effects of motherhood on the brain. Frontiers in neuroendocrinology, 53, 100740.

Response: Thank you for your valuable suggestion, we have provided some information about transition women. We have elaborated this information in the introduction. 

3. In addition, it would be helpful if the authors provided more information regarding the content of the application.

Response: Thank you for the comment. The information regarding MIESRA mHealth contents were presented in the table 1. 

4. Finally, the authors are invited to highlight strengths and weaknesses of MIESRA, but especially the future directions that the study's findings imply

Response: Thank you for the suggestion. We have revised the strengths and limitation of this study. 

“The strengths of this study include (1) the intervention utilizing mHealth media, which allows respondents to access it without limitations of time and place (2) MIESRA mHealth, an application that provides crucial information during pregnancy, pre-test and posttest assessments for knowledge, and daily activities comprising instructional videos on mindfulness, audio materials, and diary notes, (3) MIESRA mHealth is easy to install, use, and comprehend, (4) detailed research methods were employed, and (5) statistical tests conducted by experts to ensure robust results. Additionally, the availability of this application can support the limited number and time constraints of health workers in providing antenatal health services. It enables monitoring of the psychological health of pregnant women and their partners, thereby enhancing the emotional bond between the mother and the foetus.

 However, this study had several limitations, including (1) the participants were limited to a specific area. Therefore, conducting research in broader locations while considering beliefs and culture becomes essential; (2) MIESRA mHealth still had certain limitations. Researchers were unable to examine it in real-time to determine when participants engaged in mindfulness interventions and learning activities such as reading educational materials or watching educational videos. This aspect needs improvement in future work; (3) The interventions in this study were a single package; therefore, specific interventions were not conducted each week; (4) The generalization of research results is limited to pregnant women and partners without mental health disruptions. Further research is recommended for groups of patients diagnosed with mental illnesses. Overall, these limitations highlight the areas for improvement and suggest potential directions for future study.”

---

## [Decision Letter · Decision Letter 1]

11 Jul 2023

MIESRA mHealth: Marital Satisfaction During Pregnancy

PONE-D-23-01469R1

Dear Dr. Misrawati,

We’re pleased to inform you that your manuscript has been judged scientifically suitable for publication and will be formally accepted for publication once it meets all outstanding technical requirements.

Kind regards,

Giulia Ballarotto

Academic Editor

PLOS ONE

Additional Editor Comments (optional):

Reviewers' comments:

Reviewer's Responses to Questions

**Comments to the Author**

1. If the authors have adequately addressed your comments raised in a previous round of review and you feel that this manuscript is now acceptable for publication, you may indicate that here to bypass the “Comments to the Author” section, enter your conflict of interest statement in the “Confidential to Editor” section, and submit your "Accept" recommendation.

Reviewer #1: All comments have been addressed

2. Is the manuscript technically sound, and do the data support the conclusions?

Reviewer #1: Yes

3. Has the statistical analysis been performed appropriately and rigorously? 

Reviewer #1: Yes

4. Have the authors made all data underlying the findings in their manuscript fully available?

Reviewer #1: Yes

5. Is the manuscript presented in an intelligible fashion and written in standard English?

Reviewer #1: Yes

6. Review Comments to the Author

Reviewer #1: All the comments had been addressed and well justified in the following aspect:

-the information of income related to the pregnant women

-the content of experiment

-strength of MIESRA

Thank you for all the justifications.

7. PLOS authors have the option to publish the peer review history of their article (what does this mean?). If published, this will include your full peer review and any attached files.

Reviewer #1: No

---

## [Editor Report · Acceptance letter]

13 Aug 2023

PONE-D-23-01469R1 

MIESRA mHealth: Marital Satisfaction During Pregnancy 

Dear Dr. Misrawati:

I'm pleased to inform you that your manuscript has been deemed suitable for publication in PLOS ONE. Congratulations! Your manuscript is now with our production department. 

Kind regards, 

on behalf of

Dr Giulia Ballarotto 

Academic Editor

PLOS ONE